# Genomic Analysis of *Leptolyngbya boryana* CZ1 Reveals Efficient Carbon Fixation Modules

**DOI:** 10.3390/plants12183251

**Published:** 2023-09-13

**Authors:** Xiaohui Bai, Honghui Wang, Wenbin Cheng, Junjun Wang, Mengyang Ma, Haihang Hu, Zilong Song, Hongguang Ma, Yan Fan, Chenyu Du, Jingcheng Xu

**Affiliations:** 1College of Life and Environment Science, Huangshan University, Huangshan 245041, China; wjj18855852632@163.com (J.W.); mmydsg1119@163.com (M.M.); cxezrhhh1219@163.com (H.H.); sty128971@163.com (Z.S.); mhg15856731037@163.com (H.M.); wzcfdgy@163.com (Y.F.); ducy163@163.com (C.D.); 2Huangshan Institute of Product Quality Inspection, Huangshan 245000, China; ahhswhh2023@163.com; 3School of Life Sciences, University of Science and Technology of China, Hefei 230027, China; chengwb@mail.ustc.edu.cn; 4School of Life Science and Technology, Henan Institute of Science and Technology, Xinxiang 453003, China

**Keywords:** Xin’anjiang River basin, *Leptolyngbya boryana*, genome sequence, carbon fixation, CO_2_-concentrating mechanisms

## Abstract

Cyanobacteria, one of the most widespread photoautotrophic microorganisms on Earth, have evolved an inorganic CO_2_-concentrating mechanism (CCM) to adapt to a variety of habitats, especially in CO_2_-limited environments. *Leptolyngbya boryana*, a filamentous cyanobacterium, is widespread in a variety of environments and is well adapted to low-inorganic-carbon environments. However, little is currently known about the CCM of *L. boryana*, in particular its efficient carbon fixation module. In this study, we isolated and purified the cyanobacterium CZ1 from the Xin’anjiang River basin and identified it as *L. boryana* by 16S rRNA sequencing. Genome analysis revealed that *L. boryana* CZ1 contains β-carboxysome shell proteins and form 1B of Rubisco, which is classify it as belonging to the β-cyanobacteria. Further analysis revealed that *L. boryana* CZ1 employs a fine CCM involving two CO_2_ uptake systems NDH-1_3_ and NDH-1_4_, three HCO_3_^−^ transporters (SbtA, BicA, and BCT1), and two carboxysomal carbonic anhydrases. Notably, we found that NDH-1_3_ and NDH-1_4_ are located close to each other in the *L. boryana* CZ1 genome and are back-to-back with the *ccm* operon, which is a novel gene arrangement. In addition, *L. boryana* CZ1 encodes two high-affinity Na^+^/HCO_3_^−^ symporters (SbtA1 and SbtA2), three low-affinity Na^+^-dependent HCO_3_^−^ transporters (BicA1, BicA2, and BicA3), and a BCT1; it is rare for a single strain to encode all three bicarbonate transporters in such large numbers. Interestingly, *L. boryana* CZ1 also uniquely encodes two active carbonic anhydrases, CcaA1 and CcaA2, which are also rare. Taken together, all these results indicated that *L. boryana* CZ1 is more efficient at CO_2_ fixation. Moreover, compared with the reported CCM gene arrangement of cyanobacteria, the CCM-related gene distribution pattern of *L. boryana* CZ1 was completely different, indicating a novel gene organization structure. These results can enrich our understanding of the CCM-related gene arrangement of cyanobacteria, and provide data support for the subsequent improvement and increase in biomass through cyanobacterial photosynthesis.

## 1. Introduction

Since the Industrial Revolution, fossil fuels have been the backbone of the world’s energy system, driving the rapid development of modern human civilization and economic growth [1,2]. However, the increasing energy demand of human activities leads to a large amount of fossil resource consumption, triggering many problems, the most important of which are energy shortages and greenhouse effects [1,3,4]. In recent years, the warming caused by the greenhouse effect has become an environmental, scientific, political, and economic issue of global concern [5]. Of all the greenhouse gases, CO_2_ contributes the most to the greenhouse effect [6]. The concentration of CO_2_ in the atmosphere increased from 0.028% before the Industrial Revolution to 0.04%, which has caused irreversible damage to the environment [7]. At present, technologies such as reducing fossil energy use, improving energy efficiency, and using new clean energy sources are mainly used to reduce CO_2_ production or sequestration [8]. Among these, the development of stable, safe, and environmentally acceptable carbon capture and storage technology, especially microbial carbon-sequestration technology [8,9], has become a research hotspot.

Microalgae, one of the oldest photosynthetic organisms on Earth, are typical representatives of biological carbon sequestration, which can efficiently utilize light, CO_2_, and water for photosynthesis to produce O_2_ and carbohydrates and further synthesize nutrients such as fats and proteins [10]. Microalgae can fix CO_2_ from the atmosphere, industrial waste gases, and soluble carbonates through photosynthesis and store chemical energy [11], and their photosynthetic efficiency is 10–50 times higher than that of plants [11,12]. In most microalgae, photosynthetic carbon metabolism is primarily based on the C3 pathway (also known as the Calvin–Benson cycle), in which the enzyme Rubisco plays an important role in the conversion of CO_2_ into organic compounds, but the binding of Rubisco to CO_2_ is very weak due to the properties of oxygenase [13,14]. At the same time, the atmospheric O_2_ concentration is usually higher than that of CO_2_, which is not conducive to carbon sequestration. In response, microalgae have evolved CCMs to increase the CO_2_ concentration in the vicinity of Rubisco [15]. The CCMs of microalgae include: (i) inorganic carbon uptake systems; (ii) a microcompartment containing Rubisco, such as a carboxysome [16]. Interestingly, microalgae in different habitats often have different CCM components [16,17]. Therefore, an in-depth understanding of the molecular composition and gene arrangement of CCM components in microalgae can provide us with fundamental knowledge to improve the photosynthetic efficiency of these organisms or plants.

The genus *Leptolyngbya*, a filamentous cyanobacteria, is widely distributed in surface soil, freshwater, brackish water, and seawater, and is well adapted to low-inorganic-carbon environments [18]. In addition, *Leptolyngbya* has a wide range of uses such as alloprotein production [19], biological nitrogen fixation [20], green manure and nutrient regulator for rice fields [21], and the removal of heavy metals from water bodies [22]. It is worth noting that *Leptolyngbya* also has a high-affinity and high-capacity CCM, which may help it to better adapt to intense white light irradiation to avoid phototoxicity, suggesting that *Leptolyngbya*’s CCM has novel properties [23]. However, little is known about the composition and gene distribution of *Leptolyngbya*’s CCM. We successfully isolated a filamentous cyanobacterium, *Leptolyngbya boryana* CZ1, from the Xin’anjiang River basin. Genome sequencing revealed that *L. boryana* CZ1 contains a β-carboxysomal shell protein and form 1B of Rubisco. An analysis of the carboxysomal shell proteins showed that *L. boryana* CZ1 contains all carboxysome structural proteins such as CcmK3, CcmK2, CcmK1, CcmLMN, CcmO, and CcmP, except for CcmK4. However, the structures of CcmN and CcmO were significantly different from those of the reported homologs, suggesting a novel pattern of carboxysome assembly. Further analysis revealed that *L. boryana* CZ1 uses a fine CCM involving two CO_2_ uptake systems NDH-1_3_ and NDH-1_4_, three HCO_3_^−^ transporters, and two carbonic anhydrases. Compared with the reported CCM gene arrangement of cyanobacteria, the CCM-related gene distribution pattern of *L. boryana* CZ1 was completely different, indicating a novel gene organization structure. These results enrich our understanding of the CCM-related gene arrangement in cyanobacteria.

## 2. Results and Discussion

### 2.1. Isolation and Morphology of Leptolyngbya boryana CZ1

To search for algal species with a high carbon-sequestration capacity, we collected water samples from seven sampling points in the Xin’anjiang River basin every week since 2021 and isolated and purified planktonic algae from these points. The alga CZ1 was isolated from a water sample collected in November 2022. Under the optical microscope, the alga CZ1 showed slightly curved filaments, and the filaments were tightly intertwined with the colorless sheath (Figure 1A). Additionally, the filaments were composed of bright white to bright blue–green cylindrical cells separated from one to another by cross-walls.

Based on the 16S rRNA gene analysis, this algal strain is on the same evolutionary branch as *Leptolyngbya boryana* IU 594, *L. boryana* NIES-2135, *L. boryana* dg5, and *L. boryana* IAM M-101 (Figure 1B). Therefore, this algal strain was named *L. boryana* CZ1.

### 2.2. Genome Information of L. boryana CZ1

We first mapped the complete genome sequence of *L. boryana* CZ1. Our results showed that *L. boryana* CZ1 has a circular double-stranded DNA chromosome of 6,635,697 bp with a G + C content of 46.9% (Figure 2A). Prodigal V.2.6.3 [24] analyses revealed that *L. boryana* CZ1 has 6199 coding sequences (CDS), 3 sets of rRNA operons, and 61 tRNA genes (Table 1).

In addition, GeneMarkS v.4.28 [25] predicted that *L. boryana* CZ1 contains 6133 putative open reading frames (ORFs). Based on the Cluster of Orthologous Groups of proteins (COG) [26], the proteins of *L. boryana* CZ1 are mainly classified into energy production and conversion, amino acid transport and metabolism, carbohydrate transport and metabolism, inorganic ion transport and metabolism, etc. Notably, approximately 1400 proteins were annotated with unknown functions (Figure 2B), suggesting that the growth and metabolism of *L. boryana* CZ1 are unique. antiSMASH v.7.0 [27] analyses suggested the presence of 15 gene clusters, including 5 terpenes, 3 nonribosomal peptide synthetases (NRPSs), 2 NRPS-like fragments (NRPS-likes), 3 bacteriocins, and 2 lanthipeptides. Overall, the metabolic pathway of *L. boryana* CZ1 needs to be further analyzed, especially the carbon-sequestration pathway.

### 2.3. Carboxysomes of L. boryana CZ1

Previous studies revealed that cyanobacteria depend on the CO_2_-concentrating mechanism (CCM) to enhance carbon-sequestration capacity and adapt to CO_2_ limitation in their living environments [28]. Generally, the CCM of cyanobacteria consists of two parts: carboxysomes and C_i_ uptake systems [16]. Therefore, we first analyzed the carboxysomes of *L. boryana* CZ1 based on the Prokaryotic Genome Annotation Pipeline v.6.5 (PGAP) annotation [29]. After obtaining the RbcL sequence of *L. boryana* CZ1, evolutionary analysis was performed with the reported RbcL sequence [30]. Our results clearly showed that the RbcL of *L. boryana* CZ1 belongs to Form IB (Figure 3A) according to the reported classification criteria [30]. Meanwhile, previous studies reported that β-cyanobacteria contain Form IB of Rubisco with β-carboxysomes, which encode the *ccmKLMNO* operon [30]. However, only the genes *ccmK2*, *ccmK1*, *ccmL*, and *ccmM* were annotated in the genome of *L. boryana* CZ1, and the carboxysome shell proteins CcmN, CcmO, CcmK3, CcmK4, and CcmP were missing according to the PGAP annotation (Figure 3B). The genes *ccmM* (gene ID Q2T42_09005) and *ccmN* are usually located in the same operon, so it was easy to identify the potential *ccmN* (gene ID Q2T42_09010), which was annotated as a hypothetical protein. Considering that the carboxysomal shell proteins CcmK and CcmO contain bacterial microcompartment (BMC) domains, we sequentially located the potential *ccmK3* (gene ID Q2T42_03200), *ccmP* (gene ID Q2T42_04695), and *ccmO* (gene ID Q2T42_27155) genes. However, *ccmK4* was not found.

To further confirm the above potential carboxysome shell proteins, we predicted the structures of these proteins using AlphaFold2 [31] and performed structural comparison using the Dali server [32]. A Dali search revealed that *L. boryana* CZ1 CcmP (gene ID Q2T42_04695) shares a similar structure with the previously reported CcmP of *Synechococcus elongatus* PCC7942 (PDB code: 5LT5, Z-score 36.5, sequence identity 75%) and CcmP of *S. elongatus* PCC 6301 (PDB code: 4HT5, Z-score 36.4, sequence identity 75%), yielding an r.m.s.d. of 0.6 and 0.5 Å over 204 and 203 Cα atoms, respectively.

Interestingly, searching with the Dali server revealed that CcmK1 (PDB code 4LIW), CcmK2 (PDB code 2A1B), and CcmK4 (PDB code 4OX6) were the structural homologs of CcmK3 (gene ID Q2T42_03200) of *L. boryana* CZ1, with Z-scores ranging from 16.6 to 18.8. However, multiple-sequence alignment revealed that CcmK3 of *L. boryana* CZ1 was significantly different from the above CcmK proteins with an extra segment of the GDKL sequence (Figure 4A). Residues involved in the formation of hexamers [33] are conserved in CcmK3 of *L. boryana* CZ1 (Figure 4A). In addition, BLASTp was performed to find the homolog of CcmK3 of *L. boryana* CZ1. Our results showed that CcmK3 of *L. boryana* CZ1 was conserved in *Leptolyngbya* sp. and was grouped into the cd07057 sequence cluster of the bacterial microcompartment (BMC) domain [34]. The evolutionary analysis of proteins in this sequence cluster showed that CcmK3 of *L. boryana* CZ1 was closely related to CcmK of *Thermosynechococcus elongatus* (Figure 4B).

Notably, we mapped *ccmO* (gene ID Q2T42_27155) to the *ccm* operon (Figure 3B). Searching with conserved domains [34] revealed that CcmO contains an N-terminal BMC domain belonging to the CcmK protein family with an unknown functional region of approximately 100 amino acids at the C-terminus (Figure 5A). Multiple-sequence alignment confirmed that the C-terminal region of *L. boryana* CZ1 CcmO was significantly different from CcmO of *Synechocystis* sp. PCC 6803 and *S. elongatus* PCC 7942 (Figure 5B). Subsequently, the predicted structure by AlphaFold2 showed a flexible loop region at the C-terminus of CcmO (Figure 5C). A Dali search revealed that EtuB from *Clostridium kluyveri* (PDB code: 3IO0, Z-score 9.7, sequence identity 20%) shares a similar structure with the N-terminal BMC domain of *L. boryana* CZ1 CcmO with an r.m.s.d. of 2.7 Å over 229 Cα atoms, but the loop from the C-terminus of CcmO does not match (Figure 5C). Overall, these results implied that the CcmO of *L. boryana* CZ1 employs a novel structure.

In addition, a homology search of CcmN (gene ID Q2T42_09010) with the Dali server revealed that *S. elongatus* PCC 7942 CcmN (PDB code: 7D6C, Z-score 20.3, sequence identity 50%) [35] shares a similar structure with the N-terminal domain of *L. boryana* CZ1 CcmN, yielding an r.m.s.d. of 0.7 Å over 121 Cα atoms. However, the C-terminus of *L. boryana* CZ1 CcmN contains a flexible loop region with a length of approximately 110 amino acid residues (Figure 6A). To comprehensively compare the structural differences between *S. elongatus* PCC 7942 CcmN and *L. boryana* CZ1 CcmN, we also predicted the full-length protein structure of *S. elongatus* PCC 7942 CcmN using AlphaFold2. The predicted structure showed that there is also a flexible loop region at the C-terminus of *S. elongatus* PCC 7942 CcmN, which is significantly different from *L. boryana* CZ1 CcmN by comparison (Figure 6B). Moreover, multiple-sequence alignment indicated that the sequence of this region in the C-terminus of CcmN is variable (Figure 6C). Overall, all these results suggested that the interaction between *L. boryana* CZ1 CcmN and CcmM is different from that in previous reports [17,28].

Finally, we found a second *ccaA* (gene ID Q2T42_07200, named *ccaA2*) in the *L. boryana* CZ1 genome, apart from the *ccaA* (gene ID Q2T42_17920, named *ccaA1*) annotated by PGAP (Figure 3B). A Dali search revealed that both *L. boryana* CZ1 carbonic anhydrases CcaA1 and CcaA2 share a similar structure to the previously reported CcaA of *Synechocystis* sp. PCC 6803 (PDB code: 5SWC). In addition, multiple-sequence alignment confirmed that residues involved in catalysis and protein–protein interactions in *Synechocystis* sp. PCC 6803 CcaA [36] were conserved in both *L. boryana* CZ1 CcaA1 and CcaA2 carbonic anhydrases, indicating that both enzymes had catalytic activity. To our knowledge, this is the first time that two active carbonic anhydrases, CcaA1 and CcaA2, have been found in the genome of cyanobacteria according to previous reports [17,28], implying that *L. boryana* CZ1 converts HCO_3_^−^ to CO_2_ more efficiently.

Collectively, we located the genes encoding the shell proteins CcmK3, CcmP, CcmK2, CcmK1, CcmM, CcmN, and CcmO of the carboxysome, the CO_2_-fixing enzymes RbcL and RbcS, and two carbonic anhydrase CcaAs in the *L. boryana* CZ1 genome. However, the arrangement of these genes was different from those previously reported [17,28]. It is noteworthy that we found only CcmK3 with the absence of CcmK4 in the genome of *L. boryana* CZ1, suggesting a new pattern of its carboxysome assembly. In particular, for the first time to our knowledge, we identified two active carbonic anhydrases, CcaA1 and CcaA2, in the *L. boryana* CZ1 genome. Taken together, these results suggest that *L. boryana* CZ1 has adopted a novel mechanism for more efficient CO_2_ enrichment.

### 2.4. C_i_ Uptake Systems of L. boryana CZ1

Ci uptake systems, an important CCM, consist of three types of bicarbonate transporters and two types of CO_2_ uptake systems [28]. Thus, we mapped the Ci uptake systems in *L. boryana* CZ1. Interestingly, we easily found two types of CO_2_ uptake systems near the *ccm* operon, namely NDH-1_3_ (encoded by *cupA*, *ndhD3*, and *ndhF3*) and NDH-1_4_ (encoded by *cupB*, *ndhD4*, and *ndhF4*) (Figure 7A). Notably, to the best of our knowledge, this is the first report to show that NDH-1_3_ and NDH-1_4_ are located so close to one another in the genome compared to previous reports [17,28]. It is also noteworthy that the gene numbered Q2T42_08970, which encodes a hypothetical protein of unknown function, has an 8-base-pair overlap with *cupB* (Figure 7A).

In addition, the ATP-binding cassette type HCO_3_^−^ transporter BCT1, encoded by *cmpABCD*, was found upstream of the gene *ccaA2* (Figure 7A). Interestingly, a potential *sbtA1* is encoded upstream of the gene *cmpD*, while another potential *sbtA2* is removed from *ccmO* (Figure 7A). Searching with the Dali server revealed that both *L. boryana* CZ1 SbtA1 and SbtA2 share a similar structure to *Synechocystis* sp. PCC 6803 SbtA (PDB code: 7EGL) with an r.m.s.d. of 0.9 and 4.5 Å over 335 and 335 Cα atoms, respectively. Structural superposition showed that both *L. boryana* CZ1 SbtA1 and SbtA2 were well superimposed on the SbtA of *Synechocystis* sp. PCC 6803 (Figure 7B). Furthermore, multiple-sequence alignment revealed that residues involved in substrate binding in *Synechocystis* sp. PCC 6803 SbtA [37] were conserved in both *L. boryana* CZ1 SbtA1 and SbtA2, indicating that both transporters are functional (Figure 7C).

Finally, three potential low-affinity Na^+^-dependent HCO_3_^−^ transporters, namely *bicA1* (gene ID Q2T42_05125), *bicA2* (gene ID Q2T42_06210), and *bicA3* (gene ID Q2T42_19105), were located in the *L. boryana* CZ1 genome (Figure 7A). A structural comparison showed that the structures of the three transporters predicted by AlphaFold2 could be well superimposed onto each other (Figure 8A). Furthermore, a structural comparison using the Dali server revealed that the transmembrane domains of all three HCO_3_^−^ transporters BicA1, BicA2, and BicA3 share a similar structure to the transmembrane domain of *Synechocystis* sp. PCC 6803 BicA (PDB code: 6KI1) (Figure 8B), but their C-terminal STAS domain [38] differed significantly (Figure 8C). Multiple-sequence alignment revealed that residues involved in substrate binding in BicA of *Synechocystis* sp. PCC 6803 [38] were not conserved in *L. boryana* CZ1 BicA1, BicA2, and BicA3 (Figure 8D), suggesting a novel feature of substrate binding.

Collectively, two types of CO_2_ uptake systems and three types of bicarbonate transporters have been mapped in the *L. boryana* CZ1 genome. Remarkably, to the best of our knowledge, we report for the first time that NDH-1_3_ and NDH-1_4_ are located close to each other in the genome and possibly in the same operon. Another interesting finding is that SbtA1 and BCT1 are in close proximity to each other, which is also reported for the first time compared to previous studies [17,28]. Taken together, these novel features imply that *L. boryana* CZ1 is able to transport CO_2_ and HCO_3_^−^ more efficiently.

### 2.5. Gene Organization of the Carbon-Concentrating Mechanism in L. boryana CZ1

The distribution of CCM-related genes across the genome is crucial for the carbon-sequestration capacity of cyanobacteria. In previous studies, Klanchui et al. analyzed the distribution characteristics of CCM-related genes in 12 strains of alkaliphilic cyanobacteria [16]. Tang et al. investigated the distribution characteristics of CCM-related genes in 17 thermophilic cyanobacteria [17]. However, we found that the gene organization of CCM-related genes in *L. boryana* CZ1 was significantly different from that in alkaliphilic cyanobacteria and thermophilic cyanobacteria (Figure 9). The most obvious differences are as follows: (i) Ci uptake systems. All 12 alkaliphilic cyanobacteria had two CO_2_ uptake systems (NDH-1_3_ and NDH-1_4_) and BicA for HCO_3_^−^ transport, while most strains lacked the high-affinity Na^+^/HCO_3_^−^ symporter (StbA), and 8 strains lacked the HCO_3_^−^ transporter BCT1 (CmpABCD) [16]. Similarly, all 17 thermophilic cyanobacteria had two CO_2_ uptake systems (NDH-1_3_ and NDH-1_4_) and BicA for HCO_3_^−^ transport, while most strains lacked the high-affinity Na^+^/HCO_3_^−^ symporter (StbA), and 4 strains lacked the HCO_3_^−^ transporter BCT1 (CmpABCD) [17]. Interestingly, we found two CO_2_ uptake systems (NDH-1_3_ and NDH-1_4_), two high-affinity Na^+^/HCO_3_^−^ symporters (*sbtA1* and *sbtA2*), three low-affinity Na^+^-dependent HCO_3_^−^ transporters (*bicA1*, *bicA2*, and *bicA3*), and one HCO_3_^−^ transporter BCT1 (*cmpABCD*) in the *L. boryana* CZ1 genome (Figure 9). Notably, NDH-1_3_ and NDH-1_4_ were distributed in the genomes of 12 alkaliphilic cyanobacteria and 17 thermophilic cyanobacteria. In comparison, NDH-1_3_ and NDH-1_4_ are located close to each other in the *L. boryana* CZ1 genome and are found back-to-back with the *ccm* operon. This is a novel arrangement of genes and, as far as we know, the first time it has been discovered. Overall, these results suggest that *L. boryana* CZ1 is more efficient in the uptake and transport of Ci (including CO_2_ and HCO_3_^−^). (ii) Carboxysomal shell proteins. In total, 12 strains of alkaliphilic cyanobacteria expressed the complete β-carboxysomal shell proteins, while CcmK3/K4 was absent in only 2 strains, and only 3 strains expressed the carboxysomal β-CA, CcaA [16]. Among 17 thermophilic cyanobacteria, the β-carboxysomal shell protein CcmK2 was absent only in *Thermoleptolyngbya* strains, while CcmK3/K4 was absent in all *Thermostichus* and *Thermosynechococcus* strains. In addition, the carboxysomal β-CA, CcaA, was missing in all *Thermostichus* and *Thermosynechococcus* strains [17]. *L. boryana* CZ1 also expressed the complete set of β-carboxysomal shell proteins. However, we found that only *ccmK3* is present in the *L. boryana* CZ1 genome, apart from the classical organization of *ccmK3* and *ccmK4*. It is a novel discovery that *ccmK3* sits alone in the genome of cyanobacteria. Sommer et al. reported that the expression of *ccmK3* and *ccmK4* may increase the flexibility and permeability of the carboxysome shell assembly [39], suggesting that the *L. boryana* CZ1 carboxysome has a different and novel assembly mode from that previously reported [16,17]. In addition, we also found two carbonic anhydrases, *ccaA1* and *ccaA2*, which dispersed on either side of the main carboxysome locus (Figure 9). To our knowledge, we report for the first time the presence of two active carbonic anhydrases, CcaA1 and CcaA2, in the genome of cyanobacteria, indicating that *L. boryana* CZ1 has a more efficient HCO_3_^−^ conversion capacity. Overall, the distribution characteristics of the CCM-related genes in the *L. boryana* CZ1 genome (Figure 9) are different from those previously reported [17,28], indicating a new gene organization and a novel mechanism of more efficient CO_2_ fixation. This may be the reason why *Leptolyngbya* has a high-affinity and high-capacity CCM, which can better adapt to intense white light irradiation to avoid phototoxicity [23].

Taken together, the distribution characteristics of CCM-related genes in the *L. boryana* CZ1 genome (Figure 9) were different from those previously reported [17,28], suggesting a novel gene organization and a new mechanism for more efficient CO_2_ fixation. This pattern of gene organization can be generalized to other *L. boryana*, such as *L. boryana* IU 594, *L. boryana* NIES-2135, and *L. boryana* dg5 (Figure 9).

## 3. Materials and Methods

### 3.1. Isolation and Identification of an Alga CZ1

To isolate carbon-sequestration bacteria, water samples were collected weekly from seven sampling sites in the Xin’anjing River basin beginning in 2021. The collected samples were transported back to the laboratory in ice bags for processing. After filtering with eight layers of gauze to remove impurities, water samples were treated with 5.0 and 1.2 µm filter membranes. The filtered membrane was repeatedly rinsed with BG11 medium, and then the elution of BG11 medium was transferred into a conical flask containing fresh BG11 medium. The conical flask was placed at 25 °C under a light intensity of 4000 lx with a 12 h light/12 h dark cycle. After 4~5 days of incubation, the grown algal solution could be used for further isolation and purification.

One milliliter of algal solution was diluted 10-fold, and then 5 µL of algal solution was placed on a slide for single algal cell isolation under an optical microscope. The picked single algal cell was transferred into a 24-well plate containing fresh BG11 medium and grown at 25 °C under a light intensity of 4000 lx with a 12 h light/12 h dark cycle. After 3 days of incubation, 5 µL of algal solution was taken for purity checking with an optical microscope. The pure algal solution was repeatedly purified on BG11 agar plates by scraping. According to the above purification methods, more than 10 strains of pure algae were obtained from the water samples of the Xin’anjiang River basin, mainly including *Anabaena*, *Oscillatoria*, and *Aphanizomenon*, etc. One strain of filamentous algae was named CZ1.

### 3.2. Genomic DNA Extraction

A single colony of CZ1 algae was selected and cultured in BG11 liquid medium to an OD_680 nm_ of 0.6~0.8. A 200 mL algal solution of CZ1 was centrifuged at 3000 rpm at 10 °C for 10 min. The algal sediment was washed with 1× PBS buffer (10 mM Na_2_HPO_4_, 1.8 mM KH_2_PO_4_, 2.7 mM KCl, and 137 mM NaCl, pH 7.4) 2~3 times. After sufficiently grinding with liquid nitrogen, the algal sediment was resuspended in 1× lysis buffer (20 mM Tris-HCl pH 8.0, 2 mM EDTA, and 1.0% SDS) and incubated at 56 °C for 1 h with 10 mg/mL proteinase K. The suspension was subsequently treated twice by adding an equal volume of chloroform–isoamyl alcohol (24:1), and the DNA was precipitated by ice-cold isopropyl alcohol. The DNA precipitate was washed twice with 75% ethanol and then resuspended in elution buffer containing 10 mM Tris pH 8.5 and 0.1 mM EDTA.

### 3.3. Genome Sequencing

The genome sequencing work was commissioned by Tiangen Biochemical Technology (Beijing) Co., Ltd., Beijing, China. Briefly, the sequencing library of the CZ1 genomic DNA was constructed with the ligation sequencing kit (SQK-LSK110, Oxford Nanopore Technologies, Oxford, UK) and the native barcoding expansion kit (EXP-NBD104/114, Oxford Nanopore Technologies). Genome assembly was carried out by the software Canu v.2.2 [40], and Racon v.3.3.3 [41] was used to verify the assembled qualities.

### 3.4. Genome Annotation and Analysis

The putative ORFs of CZ1 were annotated using Prodigal v.2.6.3 [24] and GeneMarkS v.4.28 [25]. The function of each ORF was predicted against the nonredundant protein database using the BLASTp program in NCBI (https://www.ncbi.nlm.nih.gov/, accessed on 10 April 2023). Alternatively, the assembled genome was also submitted to the GenBank database in NCBI to annotate the location and sequence of each gene and the coding proteins in the CZ1 genome using the PGAP v.6.5 [29]. The annotated genome was submitted to Proksee v1.0.0a6 [42] to draw the circular genome map. In addition, the structural proteins were predicted using AlphaFold2 [31] combined with the Dali server [32].

### 3.5. Phylogenetic Analysis of CZ1

The 16S rDNA of CZ1 was compared with the nr/nt nucleotide collection database in NCBI using BLASTn with the default parameters. Afterwards, the 16S rDNA sequences of 22 strains of *Leptolyngbya* sp. were retrieved from the hits to perform a phylogenetic analysis according to the previously reported method [43]. The 16S rDNA sequence of *Escherichia coli* strain U 5/41 was used as an outgroup. Multiple-sequence alignment of those sequences was constructed using the ClustalW algorithm in MEGA v.7.0.26 [44]. Subsequently, the phylogenetic tree was constructed with MEGA v.7.0.26 using the maximum-likelihood method with the default parameters and a bootstrap of 1000. Finally, iTOL v.5 [45] was used to display the phylogenetic tree.

### 3.6. Data Availability

The assembled genome of *L. boryana* CZ1 has been deposited in the GenBank database with accession number CP130144 at http://www.ncbi.nlm.nih.gov/genbank/, accessed on 10 July 2023.

## 4. Conclusions, Limitations, and Future Research

In this study, we isolated and identified a strain of *L. boryana* CZ1 from the Xin’anjiang River basin. Through whole-genome sequencing and genome analysis of *L. boryana* CZ1, we found that the distribution characteristics of CCM-related genes in the *L. boryana* CZ1 genome are different from those reported in alkaliphilic and thermophilic cyanobacteria. In particular, two CO_2_ uptake systems (NDH-1_3_ and NDH-1_4_) are located close to each other and back-to-back with the *ccm* operon in the *L. boryana* CZ1 genome, indicating a novel gene arrangement. In addition, the strain CZ1 fully encodes three bicarbonate transporters in large numbers and uniquely encodes two active carbonic anhydrases, CcaA1 and CcaA2, both of which are rare. These findings help us to better understand the diversity of CCMs in cyanobacteria and the molecular mechanism of efficient carbon sequestration. At the same time, these results are based on the *L. boryana* CZ1 genome analysis and protein structure prediction, which may have some limitations due to the limitations of the prior research technique. Further studies on the structure and function of the CCM-related proteins in CZ1 should be carried out using prokaryotic expression techniques, structural biology, and proteomics, etc., in order to accurately elucidate the molecular mechanism of their efficient carbon sequestration and to provide fundamental knowledge for improving the photosynthetic efficiency of these organisms or plants in the future.

## Figures and Tables

**Figure 1 plants-12-03251-f001:**
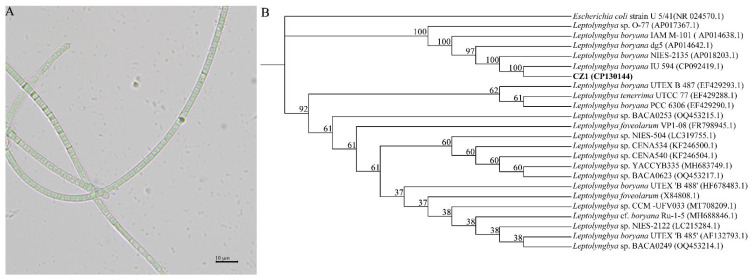
Identification of cyanobacteria CZ1. (**A**) Image of cyanobacteria CZ1 photographed by optical microscopy. (**B**) A phylogenetic tree of *Leptolyngbya boryana* CZ1 based on 16S rRNA sequences. The number in brackets for each strain is the accession number of the strain in the NCBI database.

**Figure 2 plants-12-03251-f002:**
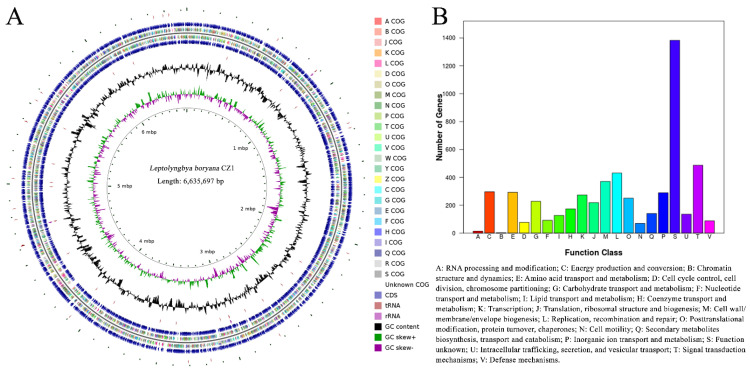
Genomic characteristics of *L. boryana* CZ1. (**A**) The chromosome map of *L. boryana* CZ1 contains four circles. From the inside to the outside, the first circle represents the GC skew (GCskew = (G − C)/(G + C)), which measures the relative content of G and C. The second circle shows the GC content. The third circle shows the classification results of tRNA, rRNA, negative chain genes, and COG genes. The fourth circle shows the results of gene COG classification, negative chain genes, rRNA, and tRNA. (**B**) Schematic representation of the COG classification of *L. boryana* CZ1.

**Figure 3 plants-12-03251-f003:**
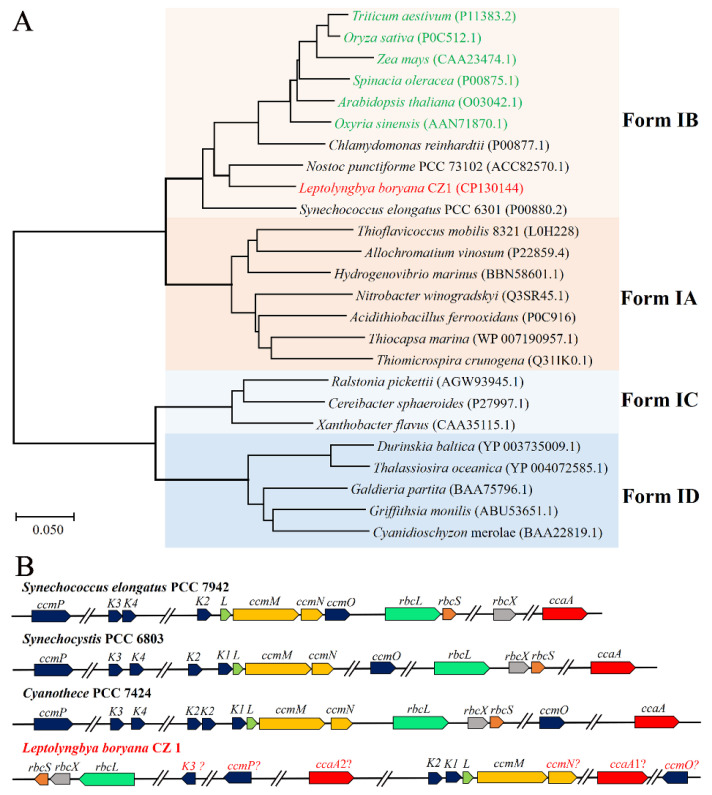
The carboxysome properties of *L. boryana* CZ1. (**A**) A phylogenetic tree of Form I Rubisco large subunits. The number in brackets for each protein is the accession number of the protein in the NCBI database. The corresponding branches of the higher plant Rubiscos are denoted in green. (**B**) Genomic structure of representative β-carboxysomal *ccm* operons according to a previous report [28]. Identical colors indicate that the genes have similar structural and/or functional products.

**Figure 4 plants-12-03251-f004:**
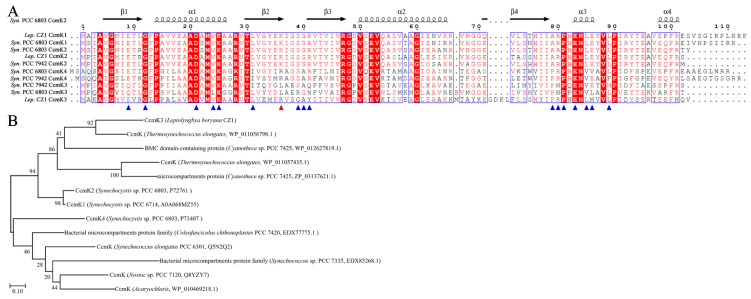
Analyses of the carboxysome shell protein CcmK3 of *L. boryana* CZ1. (**A**) Multiple-sequence alignment of *L. boryana* CZ1 CcmK3 based on proteins from *Synechococcus* sp. PCC 6803 and *S. elongatus* PCC7942. (**B**) A phylogenetic tree of the cd07057 sequence cluster of the bacterial microcompartment (BMC) domain [34].

**Figure 5 plants-12-03251-f005:**
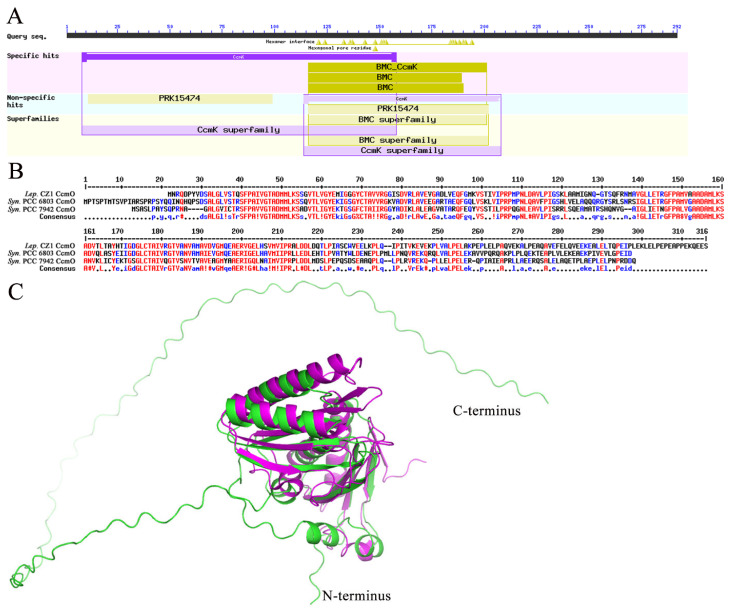
Analyses of the carboxysome shell protein CcmO of *L. boryana* CZ1. (**A**) Domain analysis of the carboxysome shell protein CcmO of *L. boryana* CZ1. (**B**) Multiple-sequence alignment of *L. boryana* CZ1 CcmO with CcmO proteins from *Synechococcus* sp. PCC 6803 and *S. elongatus* PCC7942. The same residues are shown in red, and those with similar properties are shown in blue. (**C**) Structural superposition of *L. boryana* CZ1 CcmO (green) on *C. kluyveri* EtuB (magenta, PDB code: 3IO0).

**Figure 6 plants-12-03251-f006:**
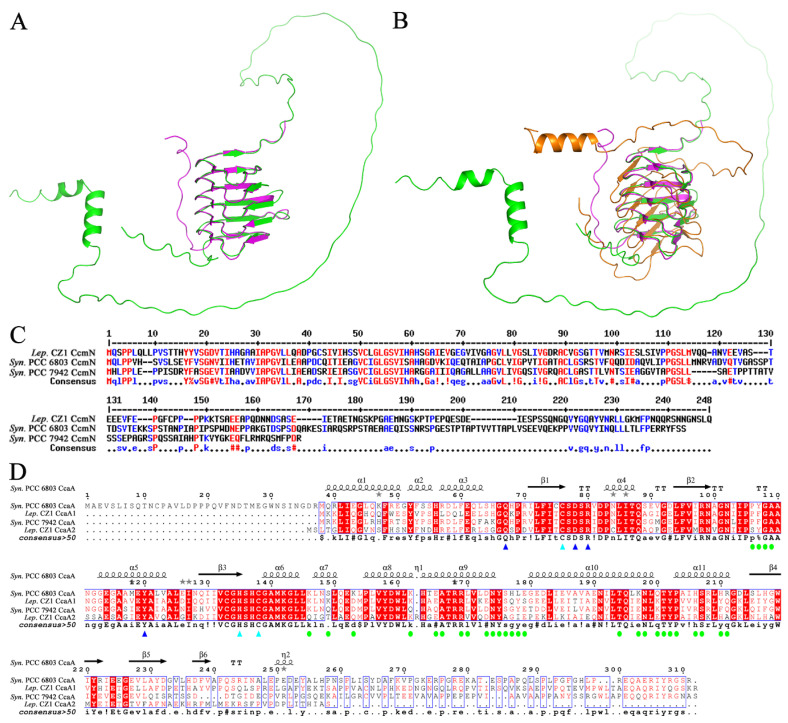
Analyses of the carboxysome shell protein CcmN and carbonic anhydrases of *L. boryana* CZ1. (**A**) Structural superposition of *L. boryana* CZ1 CcmN (green) on *S. elongatus* PCC7942 CcmN (magenta, PDB code: 7D6C). (**B**) Structural superposition of *L. boryana* CZ1 CcmN (green) on *S. elongatus* PCC7942 CcmN (magenta, PDB code: 7D6C) and the predicted full-length protein structure of *S. elongatus* PCC 7942 CcmN (orange). (**C**) Multiple-sequence alignment of *L. boryana* CZ1 CcmN with CcmN proteins from *Synechococcus* sp. PCC 6803 and *S. elongatus* PCC7942. The same residues are shown in red, and those with similar properties are shown in blue. (**D**) Multiple-sequence alignment of *L. boryana* CZ1 CcaA1 and CcaA2 with CcaA proteins from *Synechococcus* sp. PCC 6803 and *S. elongatus* PCC7942. Residues involved in catalysis are labeled with triangles, and those involved in protein–protein interactions are labeled with solid dots.

**Figure 7 plants-12-03251-f007:**
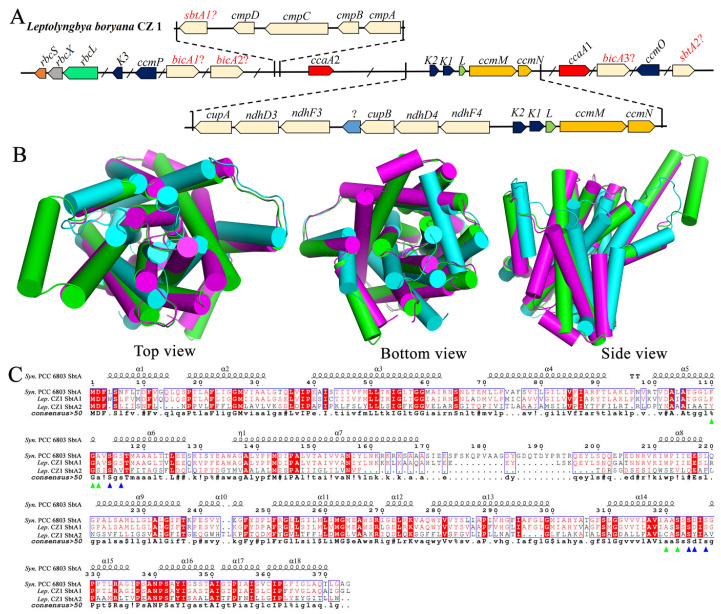
Properties of the Ci uptake systems of *L. boryana* CZ1. (**A**) Genomic organization of the Ci uptake systems of *L. boryana* CZ1. (**B**) Structural superimposition of *L. boryana* CZ1 SbtA1 (green) and SbtA2 (cyan) on *Synechococcus* sp. PCC 6803 SbtA (magenta, PDB code: 7EGL). (**C**) Multiple-sequence alignment of *L. boryana* CZ1 SbtA1 and SbtA2 with *Synechococcus* sp. PCC 6803 SbtA. Residues involved in substrate binding are labeled with triangles.

**Figure 8 plants-12-03251-f008:**
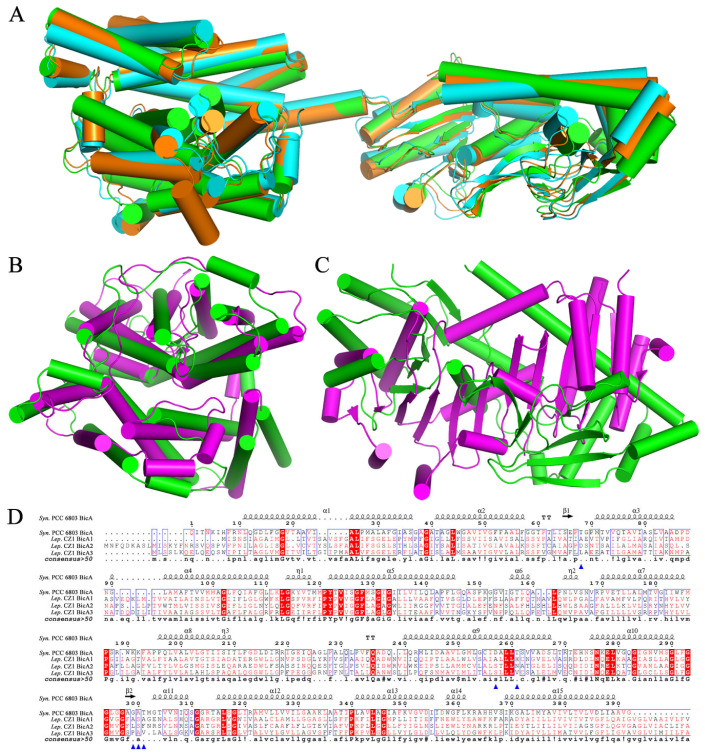
Analyses of the low-affinity Na^+^-dependent HCO3^−^ transporters of *L. boryana* CZ1. (**A**) Structural comparison of the low-affinity Na^+^-dependent HCO3^−^ transporters BicA1, BicA2, and BicA3 from *L. boryana* CZ1. (**B**) Structural superposition of the transmembrane domain of *L. boryana* CZ1 BicA1 (green) on that of *Synechococcus* sp. PCC 6803 BicA (magenta, PDB code: 6KI1). (**C**) Structure comparison of the STAS domain of *L. boryana* CZ1 BicA1 (green) on that of *Synechococcus* sp. PCC 6803 BicA (magenta, PDB code: 6KI2). (**D**) Multiple-sequence alignment of the transmembrane domain of *L. boryana* CZ1 low-affinity Na^+^-dependent HCO3^−^ transporters on that of *Synechococcus* sp. PCC 6803 BicA. Residues involved in substrate binding are labeled with triangles.

**Figure 9 plants-12-03251-f009:**
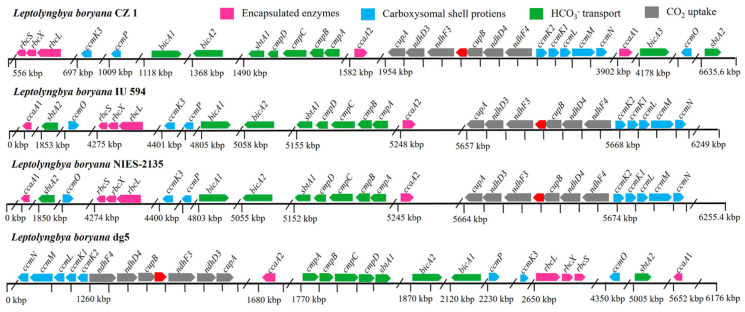
Genomic architecture of CCM-related genes in *L. boryana* CZ1. Solid arrow boxes indicate genes and transcription direction.

**Table 1 plants-12-03251-t001:** Genome information of *L. boryana* CZ1.

Features	Quantitative Value
Chromosome length (bp)	6,635,697
G + C content (%)	46.9
Total CDS	6199
Annotated CDS	6133
rRNA operon	3
tRNA genes	61

## Data Availability

Not applicable.

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
