# Peer review of "Genomic Analysis of Leptolyngbya boryana CZ1 Reveals Efficient Carbon Fixation Modules"

_plants, 2023, doi:10.3390/plants12183251_

Round 1

Reviewer 1 Report

The study titled "Genomic analysis of Leptolyngbya boryana CZ1 reveals efficient carbon fixation modules" provides valuable insights into the carbon fixation mechanisms of Leptolyngbya boryana CZ1, there are several limitations that should be considered.

The abstract lack specific information such as quantitative results and distinctive findings should be added.

Usually abbreviations are not explained at first use. So, Abbreviation must be elaborated at first use.

The study focuses on a specific strain of Leptolyngbya boryana (CZ1). And lack comparison with existing strains and even cannot be generalized for other strains or species of Cyanobacteria. So revise line 19-20 sentence and last sentence of the abstract.

Line 35 carbon sequestration capacity of plants. Check this line or revise the sentence is not clear

Introduction of the study do not convey or provide sufficient information of the topic.

Line 44 should be cited with recent studies https://doi.org/10.3390/agronomy13010269

Paragraph 1 highlight which recent and advance techniques have been adopted until now to reduce Co2 concentration.

Line 46 should be cited with recent study https://doi.org/10.1016/j.scitotenv.2022.160930

Also add the role of microorganisms in carbon sequestration.

Line 54 -60 add mechanism of the carbon sequestration by microalgae.  

Also add few lines and recent studies of the genome analysis playing role in carbon fixation modules.

The study investigates Leptolyngbya boryana CZ1 under controlled laboratory conditions. However, in natural environments, the efficiency and regulation of carbon fixation modules may differ due to various biotic and abiotic factors.

The study has not compared Leptolyngbya boryana CZ1 with other related cyanobacteria or carbon-fixing organisms. Comparative analyses with closely related species could provide additional insights into the unique aspects of carbon fixation.

The morphology and the tree is not clear. Not conveying any information.

Highlight the isolated or identified strain on the tree. Text must be clear and morphology should be also clear in figure 1.

Genes and species names should be italic in the whole MS.

Section 3.5 should be cited with recent studies DOI.10.1016/j.micpath.2020.103966

Methods should provide information how many samples were collected and on which basis CZ1 strain was used for identification.

The study also does not provide biochemical characterizations of the strain.

Add future perspective and distinctive findings of the study in conclusion.

English language need revision. I mentioned in my comments to author. also revise long sentences to clarify the confusing sentences

Reviewer 2 Report

REVIEW

Title of the paper: Genomic analysis of Leptolyngbya boryana CZ1 reveals efficient carbon fixation modules

Manuscript Number: plants-2554458

General conclusion: Major Revision.

Comments

After carefully reading the proposed paper, this paper contains an interesting proposal; my overall impression is that the manuscript presents some results that could be useful in practice. I have a good opinion about this work and recommend its acceptance after addressing the following aspects:

My comments are:

1.    The Abstract is very general. It is necessary to mention a brief description of the content of the manuscript in a clear and concise manner so that the reader can understand the content of the manuscript.

2.    All figures in the manuscript are very poor resolution, all these figures should be replaced to eps format. In addition, the caption of these figures is very long so it should be shortened.

3.    In general, it is usual that section of the introduction presents (in the following order) the topic, motivations of the work, bibliographical review, objectives, the novelty of the manuscript, and description of its sections, with no formulas, which can be moved to a section of background on the topic. This organization must be considered in the revised manuscript.

4.    The author must provide more details about the computational framework used in the manuscript. For example, software and packages used, features of the computer employed, runtimes, and other computational aspects must be added.

5.    The titles of all sub section do not take the same formats.

6.    The conclusions need to be improved. Also, the authors must add limitations to the study and more ideas for further research. Then, I suggest titling the final section as "Conclusions, limitations, and future research".

Minor editing of English language required

Round 2

Reviewer 1 Report

All comments are well revised, so MS can be accepted

Reviewer 2 Report

Thank you very much for the good response to all comments.